# The Impact of Educational Intervention on Willingness to Enroll in a Clinical Trial of a Gonorrhea Vaccine

**DOI:** 10.3390/vaccines11030648

**Published:** 2023-03-14

**Authors:** Michael Penlington, Uwe Nicolay, Ilaria Galgani

**Affiliations:** 1GlaxoSmithKline Biologicals S.A., Avenue Fleming 20, 1300 Wavre, Belgium; 2GSK Vaccines GmbH, Emil-von-Behring-Straße 76, 35041 Marburg, Germany; 3GlaxoSmithKline S.p.A., Via Fiorentina 1, 53100 Siena, Italy

**Keywords:** gonorrhea, sexually transmitted disease, vaccine trial, vaccine hesitancy, educational intervention, trial recruitment

## Abstract

Globally, >80 million new gonorrhea infections occur annually. Here, we assessed barriers to and influences on participation in a gonorrhea clinical trial and the impact of educational intervention. The survey was fielded in the US in March 2022. Higher enrollment of Black/African Americans and younger individuals than represented in the US demographic distribution reflected the higher incidence of gonorrhea in these groups. Behavioral characteristics and baseline attitudes toward vaccination were collected. Participants were probed on their knowledge of and likelihood to enroll in general and gonorrhea vaccine trials. Participants hesitant to enroll in a gonorrhea vaccine trial were given nine bullets of basic facts about the disease and asked again to rank their likelihood to enroll. Overall, 450 individuals completed the survey. Fewer participants were willing (quite/very likely) to join a gonorrhea versus a general vaccine trial (38.2% [172/450] vs. 57.8% [260/450]). The likelihood to enroll in any vaccine trial or a gonorrhea vaccine trial was greater with higher self-declared knowledge (Spearman’s ρ = 0.277 [*p* < 0.001] and 0.316 [*p* < 0.001], respectively) and baseline openness towards vaccination (*p* < 0.001 for both). Self-declared awareness of gonorrhea was associated with age (*p* = 0.001), education (*p* = 0.031), and ethnicity/race (*p* = 0.002), with older, more educated, and Black/African Americans having higher awareness. Males (*p* = 0.001) and those with more sexual partners (*p* < 0.001) were more likely to enroll in a gonorrhea vaccine trial. Educational intervention had a significant (*p* < 0.001) impact on hesitancy. Improvement in willingness to enroll in a gonorrhea vaccine trial was greatest in those initially marginally hesitant and lowest in those initially strongly hesitant. Basic educational intervention has the potential to improve recruitment into gonorrhea vaccine trials.

## 1. Introduction

Over 1 million sexually transmitted diseases (STDs) are acquired daily worldwide [1]. In 2020, the World Health Organization (WHO) estimated that there were 82.4 million new cases of gonorrhea globally [2], of which 677,769 cases were reported in the US [3]. The incidence of gonorrhea in the US has more than doubled since historic low rates were noted in 2009, making it now the second most common notifiable STD [3]. *Neisseria gonorrhoeae*, the causative agent of gonorrhea, continues to evolve and acquire resistance to multiple classes of antibiotics, leading to reduced treatment options [1,2]. Consequently, the WHO has placed *N. gonorrhoeae* on the global priority list of antimicrobial-resistant pathogens [2]. This highlights the need for new strategies for disease control [4]. Vaccination against *N. gonorrhoeae* could provide protection and help to curb the rising incidence of gonorrhea and associated drug resistance [5,6,7]. To date, however, no vaccine indicated against *N. gonorrhoeae* has been licensed.

On average, it takes 10 years from the start of phase one to the licensing of a new vaccine [8]. Moreover, the lack of surrogate markers or correlates of protection for *N. gonorrhoeae* [9] predicates a large phase three efficacy trial as part of the development process. There are currently several ongoing and planned clinical trials of vaccines to protect against gonorrhea. However, recruitment is a significant challenge. Gonorrhea vaccine clinical trials have long recruitment intervals to enroll a relatively modest number of participants (e.g., NCT04350138 [10], NCT04415424) [11]). A phase three efficacy trial for licensing could require between 2000 and 8000 participants, depending on the candidate vaccine. The recruitment of high-risk populations would be important to accelerate development. In the US, the burden of infection in men who have sex with men (MSM) is 42 times that in heterosexual men [12,13]. In addition, reported cases among Black/African Americans are 7.7 times those among the White population (548.9 per 100,000 vs. 71.1 per 100,000, respectively) [14]. The lack of awareness or availability of appropriate information may play a role in the under-representation of minority groups in clinical research in general and in studies of STDs in particular [15,16,17,18].

Although the understanding of the reasons for hesitancy toward vaccination [19,20,21,22] and the effectiveness of health education in informing vaccination decisions have advanced during the COVID-19 pandemic era [21,23,24,25], there are few studies on the potential for focused health education on enrollment decisions, especially for research on STDs. We conducted a survey among US participants examining the willingness to participate in such clinical trials based on demographic and behavioral risks, awareness of the disease, and perceptions about vaccination, and how educational intervention could facilitate informed decision-making to improve recruitment strategies in future gonorrhea vaccine trials. Here, we present the results pertaining to barriers to and motivations for taking part in a clinical study of a gonorrhea vaccine. While relevant survey results on general vaccine clinical trials have been presented here, others will be published subsequently.

## 2. Methods

### 2.1. Study Design

The 20-min, online survey consisted of 38 multiple-choice questions (mainly quantitative and some open-ended). The survey assessed participant demographic and socioeconomic characteristics, attitudes toward vaccination, knowledge, and awareness of clinical trials and STDs, and the willingness to participate in clinical trials. The full questionnaire is available in the Appendix A. The questions were designed based on regulatory guidance [26] and other publications addressing challenges with recruitment into clinical trials, particularly those including minority and high-risk populations [27,28,29,30,31,32].

The survey was designed and conducted by InSites Consulting with input from GSK and was carried out during March 2022. All participants provided electronic informed consent before the start of the survey.

### 2.2. Participant Enrollment

Participants were enrolled through an online panel recruiter who operates databases of over 1.3 million US members with representation across age, sex, and ethnicity/race in all states. Email invitations with a link to the survey were sent. Two reminders were sent to those who agreed to participate, asking them to complete the respective survey within the 2-week window. Respondents received credits upon completing a survey according to the panel’s terms and conditions, which could be exchanged for cash incentives. Participants were selected by quota sampling to facilitate sufficient representation of males and females (1:1), age groups (18–24, 25–30, and 31–45 years [2:2:1]), and ethnicities/races (Black/African American, White, and an aggregation of other ethnicities/races [2:2:1]). If more than one ethnicity/race was selected, participants were categorized as multiracial/multi-ethnic. Participants who did not consider themselves sexually active were excluded.

### 2.3. Survey Design

Enrollment of 450 participants was planned. Baseline data included age, sex, ethnicity/race, income bracket, state of residence, number of sexual partners within the previous 6 months, sexual preferences, education level, whether they had been previously vaccinated, and whether they were open to vaccination in the future (Appendix A, questions 3–12). Participants were then asked about their level of awareness about clinical research and likelihood to enroll in a vaccine clinical research study (Appendix A, question 14). All participants were asked about trusted sources of healthcare information. The study population was asked about their awareness of eight common STDs (Appendix A, question 26). Participants who may have heard of gonorrhea or confirmed having heard of the disease were probed further on their level of knowledge of the disease and the source of their knowledge (Appendix A, questions 27 and 28, respectively). All participants were asked to rank their likelihood to enroll in a gonorrhea vaccine clinical trial (Appendix A, question 29). Those who indicated that they were very unlikely (strongly hesitant) or quite unlikely (quite hesitant) to enroll were asked about barriers to enrollment. They and those who indicated that they were marginally hesitant (neither likely/unlikely) to enroll were then shown some basic education about gonorrhea (Figure 1) and asked again about the likelihood of enrolling in a gonorrhea trial (Appendix A, question 32).

Everyone who had indicated that they were quite likely or very likely to enroll, either before or after the educational intervention, was asked about influences driving their decision.

### 2.4. Data Collection and Ethics Approval

Participants reviewed the general survey and data privacy information before providing informed consent (Appendix A, question 1). All data were collected by a third party, anonymized (personally identifiable information, such as names and addresses, was not collected), and processed in compliance with the European General Data Protection Regulations and local regulations for data protection. The survey protocol (Pro00061203) was reviewed by the Advarra Institutional Review Board (IRB) and granted an exemption from IRB oversight in accordance with the criteria found in 45 CFR 46.104(d)(2).

### 2.5. Statistical Analysis

The sample size was based on the previous experience of InSites Consulting for statistical analysis of the main research question. Participants indicated their attitudes toward an item on a 5-point ordinal scale, with high numbers suggesting a high degree of agreement with the question. Rating scales were aggregated by calculating the mean rating score per item across a population. Two or more independent groups were compared with respect to an ordinal scaled variable using the Kruskal–Wallis test. Chi-square tests were used to determine whether there was a statistically significant difference between the expected and observed frequencies in one or more categories of a contingency table, assuming no association. Fisher’s exact test was additionally applied when the numbers in a contingency table were small. An adjusted residual that was higher (less) than 2.0 (−2.0) indicated that the number of cases in that cell was significantly larger (smaller) than would be expected if the null hypothesis of no association was true. Spearman’s rank correlation coefficient (ρ) measured the strength and direction of the monotonic association between two ordinal scaled variables. The null hypothesis that ρ = 0 (i.e., no association) was statistically tested. A ρ of at least ±0.40 was considered a strong, 0.30–0.39 a moderate, 0.20–0.29 a weak, and 0.01–0.19 a negligible relationship. All *p*-values were 2-sided and not adjusted for multiplicity. *p*-values of ≤0.05 were considered to be statistically significant. The Wilcoxon signed-rank test was used to assess the overall effectiveness of the educational intervention (i.e., pre–post comparison). The influence of educational intervention (Appendix A, questions 29 vs. 32) on subgroups (e.g., MSM) was analyzed only descriptively due to the small sample sizes of the subgroups.

The data were tabulated as absolute numbers, percentage of participants, and/or mean ranks. The statistical analysis was conducted using IBM SPSS Version 25.

## 3. Results

### 3.1. Participants

Of the 3411 individuals invited to participate, 493 (14.5%) completed the screening questions and fulfilled the quota requirements. A total of 450 (91.3%) individuals completed the survey and were included in at least one of the analyses (Figure 2).

The distribution of sex, age, and ethnicity/race among the survey participants generally reflected quota sampling (Table 1 and Appendix A); however, more White participants and, with the exception of Black/African Americans, fewer ethnic/racial minorities were enrolled than planned due to difficulties in recruiting the latter. Most participants (64.0% [288/450]) had had only one sexual partner in the previous 6 months (Table 1). Males and MSM had more sexual partners than females and men who did not have sex with men (Kruskal–Wallis *p* < 0.001, Appendix A). The proportion of participants in each income bracket peaked at $50,000–$74,999 (18.7% [84/450]). There was a statistically significant association between income bracket and age (chi-square *p* = 0.04), education (chi-square *p* < 0.001), ethnicity/race (chi-square *p* = 0.03), and number of sexual partners (chi-square *p* < 0.001): with higher incomes observed for older participants and university graduates, while Black/African American participants were over-represented in the low income brackets. There was a statistically significant association between ethnicity/race and prevalence of the disease in the state of residence (Kruskal–Wallis *p* = 0.025): more Black/African Americans lived in states with a higher prevalence of the disease, with White participants tending to reside in lower prevalence states. Other risk factors did not correlate with gonorrhea prevalence in the state of residence (Appendix A; Kruskal–Wallis *p* > 0.05). At baseline, 85.8% of participants (386/450) indicated that they had previously been vaccinated, and 77.8% (350/450) indicated that they were open to vaccination in the future. Openness to vaccination and previous vaccination were significantly associated with education (Kruskal–Wallis *p* < 0.001). A lower percentage of Black/African American participants than White participants (or multiracial and an aggregation of other minority ethnicities/races) were open to vaccination and had previously been vaccinated when enrolled (Table 2).

### 3.2. Willingness to Enroll in General Vaccine Clinical Trials

Of those who had been previously vaccinated, 60.4% (233/386) were either quite likely or very likely to enroll compared with 42.2% (27/64) of those who had not been previously vaccinated (mean rank: 3.73 vs. 3.22, respectively, Kruskal–Wallis *p* = 0.005). Participants who indicated that they were open to vaccination in the future were more likely (i.e., quite likely or very likely) to enroll than those who indicated that they were not open to vaccination (65.1% [228/350] vs. 32.0% [32/100]; mean rank: 3.89 vs. 2.84, Kruskal–Wallis *p* < 0.001). Participants with higher self-declared knowledge about clinical trials and higher levels of education were more likely to enroll in a clinical trial (Spearman’s ρ = 0.277, *p* < 0.001, and ρ = 0.109, *p* = 0.021, respectively; Figure 3a).

### 3.3. STD Knowledge and Awareness

Most participants had heard of all (253/450 [56.0%]) or some (191/450 [42.4%]) of the eight STDs presented to them. HIV infection was the most commonly known (92.9% [418/450]), followed by genital herpes (85.8% [386/450]) and gonorrhea (84% [376/450]). Overall, awareness of each disease was generally high (ranging from 79.8% [359/450] to 92.9% [418/450]).

Age, education, and ethnicity/race were statistically significantly associated with gonorrhea knowledge (Kruskal–Wallis *p* = 0.001, *p* = 0.031, and *p* = 0.002, respectively; Figure 4). Younger participants had lower awareness of gonorrhea, while those with higher levels of education and Black/African Americans had higher awareness. Higher gonorrhea awareness by MSM, those with more sexual partners, and males, as indicated by mean rank (Figure 4), did not attain statistical significance (Kruskal–Wallis test, *p* > 0.05).

Baseline sentiments about general vaccination did not affect awareness of gonorrhea: 84.0% (294/350) of those open to being vaccinated indicated that they had heard of gonorrhea, and 82.0% (82/100) who were not open to vaccination were also aware of gonorrhea (Kruskal–Wallis test *p* > 0.05).

### 3.4. Willingness to Enroll in a Gonorrhea Vaccine Trial

A lower percentage of the participants were very likely or quite likely to join a gonorrhea vaccine trial compared with a general vaccine clinical trial (38.2% [172/450] vs. 57.8% [260/450]). Self-declared knowledge of gonorrhea correlated with willingness to enroll (Spearman’s ρ = 0.316, *p* < 0.001; Figure 3b).

The number of sexual partners and the sex of the participants statistically significantly impacted the likelihood to enroll in a gonorrhea vaccine trial (Kruskal–Wallis *p* < 0.001 and *p* = 0.001, respectively; Figure 5), with participants with more sexual partners and men being more likely to enroll than those with fewer sexual partners and women. The higher percentage of MSM willing to enroll than of the other participants (53.1% [17/32] vs. 37.1% [155/418]) did not reach statistical significance (Kruskal–Wallis *p* > 0.05). Black/African American participants had similar overall hesitancy scores (i.e., mean ranks) to enroll in a gonorrhea vaccine trial as White participants (Figure 5). However, higher percentages of Black/African Americans were quite likely or very likely to enroll than White participants (45.8% vs. 34.7%). At baseline, 17.8% versus 30.3% of the respective populations were not open to vaccination. Of those who indicated that they were not willing to enroll, more Black/African American than White participants were very hesitant (44.2% [34/77] vs. 30.5% [47/154]), while more White than Black/African American participants were marginally hesitant (48.1% [74/154] vs. 32.5% [25/77]).

In contrast to a general vaccine clinical trial, the decision to take part in a gonorrhea vaccine trial did not appear to be affected by whether the participants had had a previous vaccination (38.6% [149/386] vs. 35.9% [23/64]; mean rank: 3.00 [*n* = 386] vs. 2.86 [*n* = 64], respectively; Kruskal–Wallis *p* > 0.05). Conversely, those who were open to vaccination were approximately twice as likely to enroll in a gonorrhea trial than those who were not open to future vaccination (42.6% [149/350] vs. 23.0% [23/100], mean ranks: 3.11 vs. 2.52, respectively; Kruskal–Wallis *p* < 0.001).

### 3.5. Change in Willingness to Enroll in a Gonorrhea Vaccine Trial following Basic Educational Intervention

Overall, 278/450 participants indicated that they had some degree of hesitancy (strongly, quite, or borderline) toward enrolling in a gonorrhea vaccine trial. After educational intervention, 32.7% (91/278) of these initially hesitant participants became less hesitant (Wilcoxon signed-rank *p* < 0.001; Figure 6), with 18.7% (52/278) indicating that they now would be quite likely or very likely to enroll. The shift was highest in people who were marginally hesitant at first and lowest in people who indicated that they were initially strongly hesitant. Of the 118 respondents who initially indicated that they would be marginally hesitant toward a gonorrhea vaccine trial, 56.8% (*n* = 67) did not change their opinion after being shown the educational intervention; 11.0% (*n* = 13) became more negative (quite or strongly hesitant), and 32.2% (*n* = 38) became positive (quite likely or very likely to participate).

The educational intervention preferentially reduced the degree of hesitancy in some higher-risk groups compared with their lower-risk counterparts. A reduction in hesitancy was observed in 40.0% (6/15) of MSM versus 32.3% (85/263) of others; 35.0% (42/120) of men versus 30.3% (47/155) of women; 39.4% (50/127) of younger participants versus 29.3% (29/99) and 23.1% (12/52) of older age groups; and 42.2% (27/64) of participants with two or more sexual partners within the previous 6 months versus 29.7% (63/212) with one or none. Conversely, the impact of the educational intervention on Black/African American participants was lower than that on White participants (23.4% [18/77] vs. 39.0% [60/154]). There was a greater impact on hesitancy by the educational intervention among those who were “open” to vaccination at the start of the survey (38.3% [77/201] vs. 18.2% [14/77] for those “not open” to vaccination).

### 3.6. Factors Influencing Willingness to Get Vaccinated and Enroll in Clinical Trials

The top trusted sources of healthcare information for all participants were their doctors (76.0% [342/450]), their families (42.4% [191/450]), and healthcare websites (40.9% [184/450]). Only 20.2% [91/450] of participants identified pharmaceutical companies as a trusted source. Social media sources (8.7% [39/450]), community religious/faith leaders (6.4% [29/450]), and members of the church/mosque (4.2% [19/450]) ranked relatively low. Black/African Americans had a lower trust than White participants in their doctor (73.2% [104/142] vs. 78.0% [184/236]) but a higher trust in their family members (50.0% [71/142] vs. 40.7% [96/236]). Overall, the most useful ways to receive information were through the research team and other healthcare professionals, with conversations (mean rank: 3.95 and 3.84, respectively) marginally preferred over emails (mean rank: 3.59 and 3.72, respectively) and an informative website about the study (mean rank: 3.77). Of those who were initially strongly or quite hesitant to enroll in any clinical trial, 45.5% (30/66) indicated that information “on how the vaccine can help society” would be essential for decision-making versus 58.5% of those who were quite or very likely to take part (152/260). On the other hand, 56.1% (37/66) and 55.4% (144/260) of the respective groups identified information on “how the vaccine may help me personally” as essential. Those who were open to vaccination had greater trust in the information they received from any source (except information from community leaders, which ranked low regardless of baseline sentiments on vaccination). This difference was highest for “government announcements” and “a conversation with the research team” (Kruskal–Wallis *p* < 0.001). Overall, 59.7% (92/154) of individuals who were very or quite hesitant to take part in a gonorrhea trial expressed anxiety about being judged by society. This compares with 34.5% (29/72) of those who were quite or very hesitant to take part in a general clinical trial.

A higher percentage of Black/African American participants than White participants who indicated any degree (very, quite, and borderline) of hesitancy to enroll in a gonorrhea clinical trial mistrusted pharmaceutical companies (70.0% [35/50] vs. 51.9% [40/77]) and had concerns about mistreatment and misuse of personal information (70.0% [35/50] vs. 59.7% [46/77]).

Of the 61.6% (277/450) of survey participants who had “definitely heard” of gonorrhea and indicated that they knew at least something about the disease, the school/college nurse (34.7% [96/277]) was the most frequently selected source of information, followed by an official medical website (27.4% [76/277]), and then a health information leaflet, brochure, or campaign (26.7% [74/277]). Family, friends, and partners (range 9.0 [25/277]–13.0% [36/277]) were less frequently identified as sources of information about gonorrhea.

## 4. Discussion

Gonorrhea is a major public health priority [2], for which vaccines can offer the only sustainable solution for control [34]. The ability to preferentially enroll high-risk populations in clinical efficacy trials is essential to expedite the licensure pathway. This study found that basic medical education significantly reduced hesitancy to participate in a gonorrhea vaccine trial. There was a greater reduction in hesitancy among more sexually active participants and MSM compared with their lower-risk counterparts. There was also a higher impact on male participants than on female participants, and on younger participants than on older participants. Males and younger members (20–24 years of age) of the US population have higher gonorrhea rates than females or older individuals [14], although not to the same extent as those associated with high-risk sexual behavior. Conversely, the reduction in hesitancy among Black/African American participants following the educational intervention was lower than that among White participants, although the gonorrhea incidence rate was 7.7-fold higher [14]. This could reflect the identification in the medical education material of sexual behavior but not demographic groups, such as Black/African Americans, as a risk factor. Almost twice the proportion of Black/African Americans than White participants were not open to vaccination when faced with a binary choice at the start of the survey. In addition, a higher percentage of Black/African Americans than White participants were strongly hesitant. If, as previously suggested [20,35], the anxiety and mistrust in clinical research were deeply rooted, this may be more complex to address through communication strategies.

The finding that educational intervention had the greatest impact on those who were marginally hesitant and the least impact on those who were strongly hesitant is perhaps not surprising when considering the higher level of mistrust driving greater hesitancy [36]. However, the result is in contrast to another recent survey on attitudes toward the COVID-19 vaccination, which found that educational intervention significantly impacted only those who were strongly hesitant [25]. The discrepancy could have been due to the much lower general awareness of gonorrhea, making medical education more effective, except for those with strong negative sentiments against vaccinations. The results presented here agree with Freeman et al. [25] that information on the personal advantages of vaccination, rather than the collective benefit, has the greatest influence on hesitant populations. Therefore, decision-making becomes more ego-centric when the perception of risk from vaccines is greatest. Common concerns selected by participants who were strongly or quite hesitant to enroll in a gonorrhea clinical trial, such as risks and benefits of the vaccine, vaccine safety, and a lack of relevant information about the disease and/or the clinical trial, offer an opportunity for communicational intervention to inform decision-making. Ultimately, the amount of investment required, the content of the medical educational material, and the effectiveness of a communication strategy will be impacted by the causes of the hesitancy [21,36,37,38,39,40].

As shown here, the relationship between awareness and willingness to participate in clinical research is complex. Participants with higher levels of education had a higher awareness of clinical research and gonorrhea. However, educational levels statistically significantly correlated only with the willingness to take part in a general vaccine trial but not a gonorrhea clinical trial. Perceivably, some participants with greater levels of gonorrhea health literacy realized that they were at lower risk, and therefore they were less likely to enroll. Nonetheless, a major challenge is that both health literacy and willingness to participate in clinical research are partly driven by socioeconomic status rather than disease risk [41]. Various socioeconomic, political, and demographic factors may contribute toward vaccine hesitancy in general [19,20,41,42]. Stigma around STDs [43,44] is an additional challenge for recruitment in a gonorrhea clinical trial. Lower socioeconomic status may reflect generally poorer health literacy levels, reduced health status, and higher-risk behaviors (e.g., smoking) [45]. Nonetheless, the finding that insufficient knowledge was a more commonly identified concern than trust among strongly or quite hesitant individuals highlights the potential for relatively light informational intervention to have a significant impact. This study found that low awareness of the clinical trial process in general and of gonorrhea specifically was associated with a reluctance to engage in clinical research. Improvement in health literacy, preferentially through healthcare and other professionals, is therefore essential to inform decisions on enrollment in clinical trials.

In the US, Black/African Americans have both lower education levels (27.8% vs. 37.5% of the respective ≥25-year-old populations had at least a bachelor’s degree [46]) and a higher number of gonorrhea cases (over 7-fold higher [14]) than the White population. Among Black/African Americans who were hesitant to enroll in a gonorrhea vaccine trial, levels of trust in pharmaceutical companies and vaccine research were lower than for their White counterparts, while concerns about mistreatment and misuse of personal data were higher. Although deep-rooted anxiety and distrust of clinical research are more complex to address through communication strategies, medical education may be more effective where there is a greater disparity between the risk of infection by gonorrhea and awareness of the disease.

A limitation of this survey is that educational intervention was applied only to participants who showed some degree of hesitancy. It cannot be ruled out that for some participants, the improvement in willingness to enroll in a gonorrhea trial was due to chance, irrespective of the educational intervention—a consequence of regression to the mean. This appears to be a contributory factor, as a proportion of those who were borderline hesitant (i.e., category 3 on the 1–5 hesitancy scale) increased hesitancy after educational intervention, although there was a >3-fold greater shift toward reduced hesitancy in this group after the intervention. The relatively limited sample size of 450 participants meant that only one educational intervention could be assessed. The information was therefore designed to be short and easily digestible, and, aside from informing that gonorrhea is transmitted through unprotected sex, it did not explicitly discuss the demographic and other baseline considerations that lead to higher risk. It should also be noted that, while hypothetical willingness has been shown to be statistically associated with actual enrollment, not all individuals who indicate a willingness to enroll would necessarily do so [47].

Statistical analysis on the effectiveness of the educational intervention on subsets, including those at higher risk, was descriptive in nature due to the limited numbers of participants in these subsets and challenges with multiplicity. The influence of social determinants of health (SDOH) was not investigated for the same reason. Nonetheless, a more profound appreciation of the influence of SDOH on willingness to engage in clinical research would enable more effective content for communicational intervention.

Black/African Americans were over-represented versus the US population (32% vs. 14% of the US population), and White participants were under-represented (52% vs. 76% of the US population) [43], to reflect the prevalence of gonorrhea and vaccine hesitancy [14,20] in the respective populations. Slightly more White individuals than planned were enrolled (236 enrolled vs. 225 planned) due to challenges in recruiting minority ethnicities/races other than Black/African Americans. Therefore, the main ethnicity/race assessments on hesitancy were between the relatively higher-risk Black/African American and the lower-risk White populations. We did not investigate the reasons why some ethnic/racial minorities were more reluctant to participate in this survey; however, they may reflect general low engagement in clinical research and healthcare, as previously described by others [22,35]. The FDA guidance to the industry promotes enrollment practices that would lead to clinical trials that better reflect the indicated population [26]. Some suggestions, such as including online/social media recruitment strategies to identify participants, may also be relevant to survey recruitment for better representation of minorities. As only a few individuals indicated that their sex at birth was non-binary/preferred not to say, they were not sufficient for a meaningful analysis. Other demographic and baseline characteristics broadly reflected those of the US population. A total of 30.9% of the population had bachelor’s degrees or higher, which is similar to 32.9% for the US average, although the 2021 census data excluded individuals younger than 25 years old. Similarly, over two-thirds of participants indicated that they were open to vaccines, which is broadly consistent with published accounts across the US and around the world [25,36,44].

To our knowledge, this is the first quantitative survey to investigate the relationship between risk, knowledge, and willingness to enroll in a gonorrhea vaccine study. Educational intervention reduced the degree of hesitancy in almost a third of those initially unwilling to enroll in a gonorrhea trial. Strong baseline sentiments, personal perceptions of risk, and attitudes against vaccination may have diluted the impact of the intervention. Nevertheless, a basic educational intervention, delivered through trusted healthcare professionals to high-risk communities, has the potential to significantly improve the recruitment of at-risk individuals into vaccine clinical trials for STDs.

## Figures and Tables

**Figure 1 vaccines-11-00648-f001:**
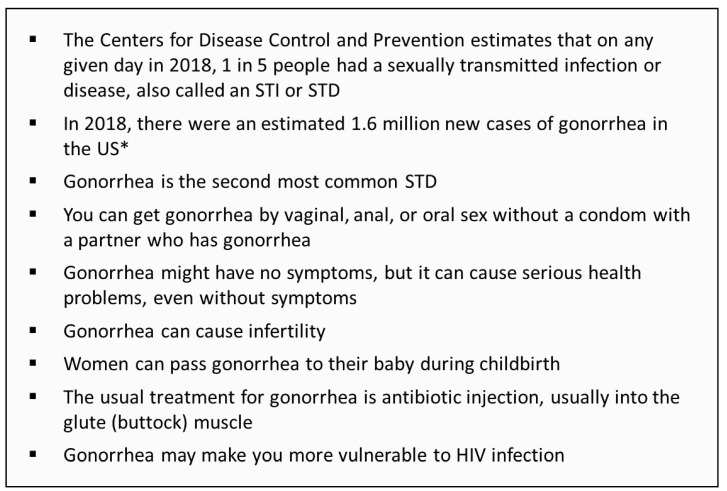
Basic educational information about gonorrhea given to participants who report that they would be neutral, quite unlikely, or very unlikely to participate in a gonorrhea vaccine clinical trial. * Incidence is the estimated number of new infections—diagnosed and undiagnosed [33]. HIV, human immunodeficiency virus; STD, sexually transmitted disease; STI, sexually transmitted infection.

**Figure 2 vaccines-11-00648-f002:**
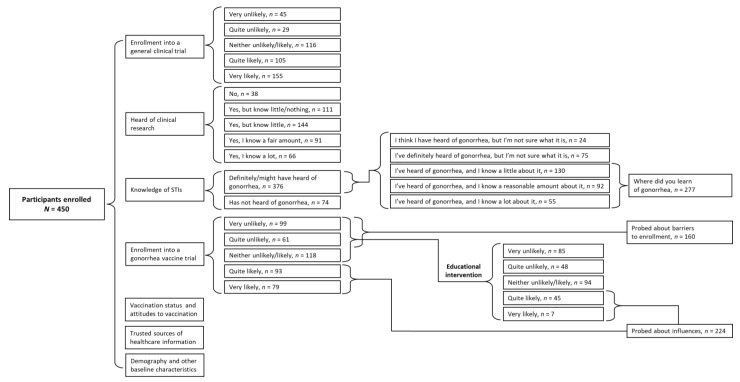
Dataset analyzed in this survey.

**Figure 3 vaccines-11-00648-f003:**
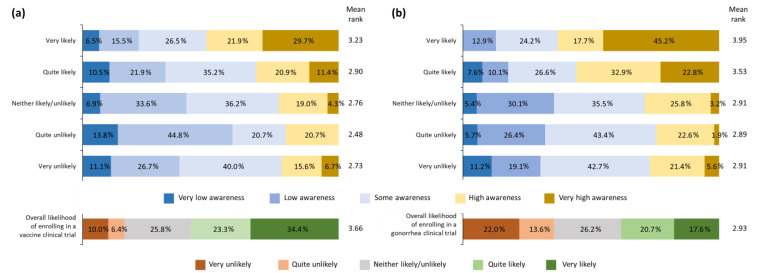
Willingness to enroll in (**a**) a general clinical research trial or (**b**) a gonorrhea vaccine trial according to self-declared knowledge. Correlation between self-declared knowledge and willingness to enroll in a general vaccine clinical trial: Spearman’s ρ = 0.277, *p* < 0.001, and a gonorrhea vaccine trial: Spearman’s ρ = 0.316, *p* < 0.001. Very unlikely = I would be very unlikely to consider taking part; quite unlikely = I would be quite unlikely to consider taking part; neither likely/unlikely = I would neither be likely nor unlikely to consider taking part; quite likely = I would be quite likely to consider taking part; very likely = I would be very likely to consider taking part. Very low awareness = I think I have heard of clinical research studies/gonorrhea, but I’m not sure what they are/it is; low awareness = I have definitely heard of clinical research studies/gonorrhea, but I’m not sure what they are/it is; some awareness = I have heard of clinical research studies/gonorrhea, and I think I know a little about them/it; quite high awareness = I have heard of clinical research studies/gonorrhea, and I think I know a reasonable amount about them/it; very high awareness = I have heard of clinical research studies/gonorrhea, and think I know a lot about them/it.

**Figure 4 vaccines-11-00648-f004:**
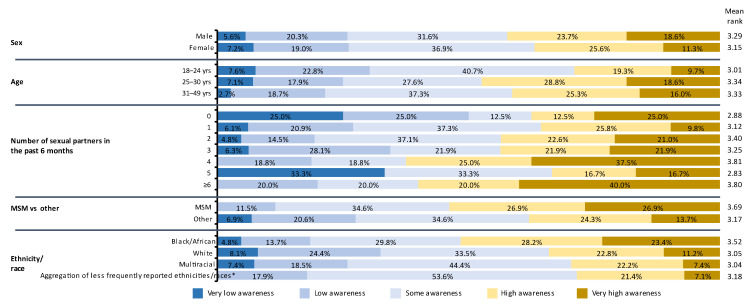
Self-declared knowledge about gonorrhea according to sex, age, number of sexual partners in the previous 6 months, MSM versus other, and ethnicity/race; (*N* = 450). Age and ethnicity/race were significantly associated with gonorrhea knowledge (Kruskal–Wallis *p* = 0.001 and *p* = 0.002, respectively). Very low awareness = I think I have heard of gonorrhea, but I’m not sure what it is; low awareness = I have definitely heard of gonorrhea, but I’m not sure what it is; some awareness = I have heard of gonorrhea, and I think I know a little about it; quite high awareness = I have heard of gonorrhea, and I think I know a reasonable amount about it; very high awareness = I have heard of gonorrhea, and I think I know a lot about it. MSM, men who have sex with men. * Less frequently reported ethnicities/races = American Indian, Alaskan Native, Asian, Native Hawaiian, Pacific Islander, Hispanic, Latino, and other ethnicities/races.

**Figure 5 vaccines-11-00648-f005:**
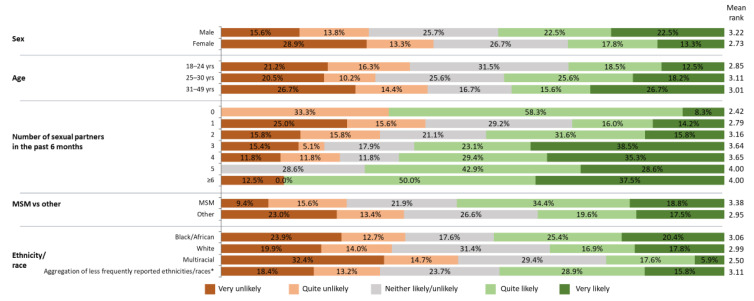
Willingness to participate in a gonorrhea vaccine clinical trial according to sex, age, number of sexual partners in the previous 6 months, MSM versus other, and ethnicity/race; (*N* = 450). The number of sexual partners and sex were significantly associated with the likelihood to enroll in a gonorrhea vaccine trial (Kruskal–Wallis *p* < 0.001 and *p* = 0.001, respectively). Very unlikely = I would be very unlikely to consider taking part; quite unlikely = I would be quite unlikely to consider taking part; neither likely/unlikely = I would neither be likely nor unlikely to consider taking part; quite likely = I would be quite likely to consider taking part; very likely = I would be very likely to consider taking part. MSM, men who have sex with men. * Less frequently reported ethnicities/races = American Indian, Alaskan Native, Asian, Native Hawaiian, Pacific Islander, Hispanic, Latino, and other ethnicities/races.

**Figure 6 vaccines-11-00648-f006:**
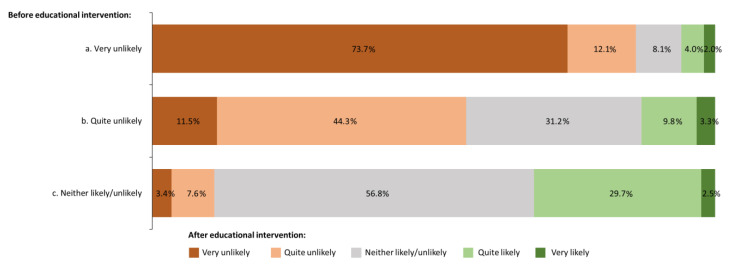
Willingness to enroll in a gonorrhea vaccine clinical trial after reviewing educational information according to those initially selecting (**a**) very likely (*n* = 99), (**b**) quite likely (*n* = 61), and (**c**) neither likely/unlikely (*n* = 118). Difference in post-pre educational intervention: −2 (1.4%), −1 (5.8%), 0 (60.1%), 1 (23.7%), 2 (6.1%), 3 (2.2%), and 4 (0.7%); Wilcoxon signed rank test *p* < 0.001.

**Table 1 vaccines-11-00648-t001:** Participant characteristics (*N* = 450).

Characteristic	Participants, *n* (%)
**Sex**	
Male	218 (48.4)
Female	225 (50.0)
Non-binary/prefer not to say	7 (1.6)
**Age group (years)**	
18–24	184 (40.9)
25–30	176 (39.1)
31–45	90 (20.0)
**Ethnicity/race**	
White	236 (52.4)
Black/African American	142 (31.6)
Mixed race	34 (7.6)
Aggregation of less frequently reported ethnicities/races *	38 (8.4)
**MSM**	
MSM	32 (7.1)
Other	418 (92.9)
**Sexual partners in the past 6 months**	
0	12 (2.7)
1	288 (64.0)
2	76 (16.9)
≥3 ^†^	71 (15.8)
Prefer not to say	3 (0.7)
**Educational level**	
College graduate	308 (68.4)
Non-college graduate	139 (30.9)
Prefer not to say	3 (0.7)
**Income ($)**	
Under $15,000 $15,000–$24,999 $25,000–$34,999 $35,000–$49,999 $50,000–$74,999 $75,000–$99,999 $100,000–$149,999 $150,000–$199,999 $200,000 and over Prefer not to say	54 (12.0) 42 (9.3) 73 (16.2) 57 (12.7) 84 (18.7) 49 (10.9) 57 (12.7) 22 (4.9) 8 (1.8) 4 (0.9)

* Less frequently reported ethnicities/races = American Indian, Alaskan Native, Asian, Native Hawaiian, Pacific Islander, Hispanic, Latino, and other ethnicities/races; ^†^ Participants with 3, 4, 5, and ≥6 sexual partners are summarized together here, but were analyzed separately and are presented by sex, age, and ethnicity/race in Appendix A. MSM, men who have sex with men.

**Table 2 vaccines-11-00648-t002:** Baseline attitudes to vaccination and vaccination status of participants by ethnicity/race.

Ethnicity/Race	Open to Vaccination (%) *n* = 350	Not open to Vaccination (%) *n* = 100	Vaccinated (%) *n* = 386	Not Vaccinated (%) *n* = 64
Black/African American	69.7	30.3	79.6	20.4
White	82.2	17.8	88.6	11.4
Multiracial/multi-ethnic	79.4	20.6	88.2	11.8
Aggregation of less frequently reported ethnicities/races *	78.9	21.1	89.5	10.5

* Less frequently reported ethnicities/races = American Indian, Alaskan Native, Asian, Native Hawaiian, Pacific Islander, Hispanic, Latino, and other ethnicities/races.

## Data Availability

Further data are published in the Appendix A, and data relating to attitudes toward general vaccine clinical trials will be published subsequently.

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
