# Peer review of "The Impact of Educational Intervention on Willingness to Enroll in a Clinical Trial of a Gonorrhea Vaccine"

_vaccines, 2023, doi:10.3390/vaccines11030648_

Round 1
Reviewer 1 Report
This is a very well-written and organized manuscript. Incredibly relevant and a good contribution to the literature. I have minor suggested revisions to further improve the submission.
1. Willingness often result in actual participation. This has been noted in HIV vaccine trial research and seems missing as a limitation here. See Buchbinder et al.
2. Using "other" as a demographic reference group is problematic. People are not "other". Please remove all references to "other"
3. Consider identifying what difficulties you faced in recruiting your indigenous and people of color group. What might be done to mitigate these challenges/overcome these barriers in the future? Given the existing disparities among these racial groups, simply stating that there were difficulties recruiting them seems unacceptable.
4. Given the challenges that BIPOC individuals face in medical settings, it would be helpful to note any racial differences in trusted sources of healthcare information.
5. Describing Black/African Americans as having deep-rooted anxieties is pathologizing. What about institutions/organizations working to build reputations of trustworthiness or as being seen as untrustworthy. The literature largely supports this as the reality and not what you have represented here.
6. Your discussion largely overlooks or ignores other significant social determinants of health needs (in addition to SES) in underserviced populations as well as provider bias (who is made aware and referred to existing trials) and access to other resources.
Author Response
Penlington M, et al. The impact of educational intervention on willingness to enroll into a clinical trial of a gonorrhea vaccine
Response to reviewers
|
Comment |
Response |
|
Reviewer 1 |
|
|
This is a very well-written and organized manuscript. Incredibly relevant and a good contribution to the literature. I have minor suggested revisions to further improve the submission. |
Many thanks for your positive feedback. |
|
1. Willingness often result in actual participation. This has been noted in HIV vaccine trial research and seems missing as a limitation here. See Buchbinder et al. |
Thank you for the suggestion. The following text (Line 469) has been added: It should also be noted that while hypothetical willingness has been shown to be statistically associated with actual enrollment, not all individuals who indicate willingness to enroll would necessarily do so [45]
|
|
2. Using "other" as a demographic reference group is problematic. People are not "other". Please remove all references to "other" |
Throughout the text, figures and supplementary material, the following terms have been amended: · ‘Non-binary/other’ has been updated to ‘Non-binary/prefer not to say’ · ‘Other’ in relation to MSM has been updated to ‘Men who do not have sex with men’ · Other in relation to ethnicity/race category has been updated to ‘Aggregation of less frequently reported ethnicities/races’. The footnotes for Tables 1 and 2 and Figures 4 and 5 have been updated to ‘* Less frequently reported ethnicities/races = American Indian, Alaskan Native, Asian, Native Hawaiian, Pacific Islander, Hispanic, Latino, other ethnicities/races’ |
|
3. Consider identifying what difficulties you faced in recruiting your indigenous and people of color group. What might be done to mitigate these challenges/overcome these barriers in the future? Given the existing disparities among these racial groups, simply stating that there were difficulties recruiting them seems unacceptable.
|
We have included the following text in the discussion (Line 493):
We did not investigate why some ethnic/racial minorities were more reluctant to participate in this survey, however they may reflect general low engagement in clinical research and healthcare, previously described by others [22, 35]. The FDA guidance to industry promotes enrollment practices that would lead to clinical trials that better reflect the indicated population [26]. Some suggestions, such as including online/social media recruitment strategies to identify participants, may also be relevant to survey recruitment for better representation of minorities.
|
|
4. Given the challenges that BIPOC individuals face in medical settings, it would be helpful to note any racial differences in trusted sources of healthcare information. |
We have included the following text in the discussion (Line 349):
Black/African Americans had a lower trust than White participants in their doctor (73.2% [104/142] vs 78.0% [184/104]) but a higher trust in their family members (50.0% [71/142] vs 40.7% [96/236]).
|
|
5. Describing Black/African Americans as having deep-rooted anxieties is pathologizing. What about institutions/organizations working to build reputations of trustworthiness or as being seen as untrustworthy. The literature largely supports this as the reality and not what you have represented here. |
We have updated the paragraph in the discussion to note the following (Line 395):
Almost twice the proportion of Black/African Americans than White participants were not open to vaccination when faced with a binary choice at the start of the survey. In addition, a higher percentage of Black/African Americans than of White participants were strongly hesitant. If, as previously suggested [20,35], the anxiety and mistrust in clinical research were deep-rooted, this may be more complex to address through communication strategies”
|
|
6. Your discussion largely overlooks or ignores other significant social determinants of health needs (in addition to SES) in underserviced populations as well as provider bias (who is made aware and referred to existing trials) and access to other resources.
|
Due to small sample size, we are unable to assess these data by SES, and other SDOH, therefore we have included a limitation in the discussion (Line 473):
Statistical analysis on the effectiveness of the educational intervention on subsets, including those at higher risk, was descriptive in nature due to the limited numbers of participants in these subsets and challenges with multiplicity. The influence of social determinants of health (SDOH) was not investigated for the same reason. Nonetheless, a more profound appreciation of the influences of SDOH on willingness to engage into clinical research would enable more effective content for communicational intervention. |
Reviewer 2 Report
This study addresses an important issue related to the impact of educational intervention on the willingness to enroll into a clinical trial of a gonorrehea vaccine. I have some observations and questions that could help to better understand the study design.
Abstract:
· Please include some details about participants characteristics (ie. Adult population, men and women, at high or low risk of STD, how were they selected to participate in the survey, how many were initially contacted, etc). This is important, because there are some high-risk population for this disease and, as it is showed in the methods section, the authors included an adequate rationale for the study population.
· They mention that self-declared awareness of gonorrhea was associated with ethnicity/race, but it is not clear how race influenced this awareness, although there is data given in the introduction that shows that the burden of infention is greater in Black/African Americans; but from this data we cannot extrapolate that the awareness will follow this trend.
· The educational intervention should be briefly described in the abstract (how and what was the information given to participants?)
Introduction:
· The authors mention that “Results relating to attitudes toward general vaccine clinical trials will be published subsequently”; however, they do make comparisons in the present study between general vaccines vs gonorrhea. At least some information should be given about general vaccines results, since that information was part of the same survey.
Methods section
· Please inform how the survey was pre-tested.
· Regarding the educational information given to participants, please verify what figure is correct (1.6 million new cases of gonorrehea in the US in 2018 or 677,769 cases reported in the US according to the WHO). Although the later was reported in 2020, very unlikely to have these differences in number of new cases.
Results section:
· Lines 169-170. “There was a statistically significant association between ethnicity/race and prevalence of the disease in the State of residence with White participants tending to reside in lower prevalence States”. Maybe it would be easier to say that more Black/African Americans live in States with higher prevalence of the disease, instead of the significant association between ethnicity/race….
· Line 213: “However, awareness of each disease was generally high”. It is not clear why the say “however”, considering the relatively high knowledge that participants declare.
· Line 215. “Age, education, and ethnicity/race were statistically significantly associated with gonorrhea knowledge”; once again the statiscally significantly association with ethnicity/race is not obvious (which race is associated with higher knowledge?). This is clear looking at the Figure, but not by reading the text.
Author Response
Penlington M, et al. The impact of educational intervention on willingness to enroll into a clinical trial of a gonorrhea vaccine
Response to reviewers
|
Comment |
Response |
|
Reviewer 2 |
|
|
This study addresses an important issue related to the impact of educational intervention on the willingness to enroll into a clinical trial of a gonorrehea vaccine. I have some observations and questions that could help to better understand the study design. |
Many thanks for your comments and suggestions. |
|
Abstract: · Please include some details about participants characteristics (ie. Adult population, men and women, at high or low risk of STD, how were they selected to participate in the survey, how many were initially contacted, etc). This is important, because there are some high-risk population for this disease and, as it is showed in the methods section, the authors included an adequate rationale for the study population. · They mention that self-declared awareness of gonorrhea was associated with ethnicity/race, but it is not clear how race influenced this awareness, although there is data given in the introduction that shows that the burden of infention is greater in Black/African Americans; but from this data we cannot extrapolate that the awareness will follow this trend. · The educational intervention should be briefly described in the abstract (how and what was the information given to participants?)
|
Please note that due to limitations of the abstract word count, we have implemented as follows: - Line 11: Higher enrollment of Black/African Americans and younger individuals than represented in US demographic distribution reflected the higher incidence risk of gonorrhea in these groups. - Line 16: Participants hesitant to enroll into a gonorrhea vaccine trial were given nine bullets of basic facts about the disease and asked again to rank their likelihood to enroll. - Line 22 Self-declared awareness of gonorrhea was associated with age (p = 0.001), education (p = 0.031), and ethnicity/race (p = 0.002); with older, more educated and Black/African Americans having higher awareness. |
|
Introduction: · The authors mention that “Results relating to attitudes toward general vaccine clinical trials will be published subsequently”; however, they do make comparisons in the present study between general vaccines vs gonorrhea. At least some information should be given about general vaccines results, since that information was part of the same survey.
|
As the relevant results from the general clinical trial that help understand the barriers and influences into gonorrhoea trail enrollment have been presented (Table 3, result section 3.2). We propose to clarify in the introduction as follows:
Line 70: Here, we present the results pertaining to barriers and motivations for taking part in a clinical study of a gonorrhea vaccine. While relevant survey results on general vaccine clinical trials have been presented here, others will be published subsequently.
|
|
Methods section · Please inform how the survey was pre-tested.
· Regarding the educational information given to participants, please verify what figure is correct (1.6 million new cases of gonorrehea in the US in 2018 or 677,769 cases reported in the US according to the WHO). Although the later was reported in 2020, very unlikely to have these differences in number of new cases.
|
The survey was tested for plain language and readability score. The survey was reviewed by internal departments and questions were based on similar surveys conducted by Insites and GSK. The survey was also reviewed by an IRB. As such there was no pre-test with members of the public. This may be something we might consider in the future.
The statement in the educational intervention that ”in 2018, there were an estimated 1.6 million new cases of gonorrhea in the US” was extracted from the CDC [At A Glance (cdc.gov)]. The WHO estimate referenced is based on actual cases reported to the CDC in 2020 while the CDC 2018 estimate referenced in Figure 1 is based on both unreported and reported cases. Hence the apparent discrepancy. We have added an asterisk to the second bullet in Figure 1 and a referenced footnote stating:
* Incidence is the estimated number of new infections – diagnosed and undiagnosed [33]. |
|
Results section: · Lines 169-170. “There was a statistically significant association between ethnicity/race and prevalence of the disease in the State of residence with White participants tending to reside in lower prevalence States”. Maybe it would be easier to say that more Black/African Americans live in States with higher prevalence of the disease, instead of the significant association between ethnicity/race….
· Line 213: “However, awareness of each disease was generally high”. It is not clear why the say “however”, considering the relatively high knowledge that participants declare.
· Line 215. “Age, education, and ethnicity/race were statistically significantly associated with gonorrhea knowledge”; once again the statiscally significantly association with ethnicity/race is not obvious (which race is associated with higher knowledge?). This is clear looking at the Figure, but not by reading the text.
|
· Many thanks for the suggestion – we have updated the text (line 180) as follows: There was a statistically significant association between ethnicity/race and prevalence of the disease in the State of residence (Kruskal-Wallis p = 0.025): more Black/African Americans lived in States with a higher prevalence of the disease, with White participants tending to reside in lower prevalence States. · We have amended the text (line 248) to say ‘Overall, awareness of each disease was generally high…’
· All mean ranks to explain the result have now been added to Figure 4. In addition, for clarity, we have added an explanation to the abstract as suggested by this reviewer and in Line 252 inserted: Younger participants had lower awareness of gonorrhea, while those with higher levels of education and Black/African Americans had higher awareness. |